# Evaluation of the Deterioration of Untreated Commercial Polystyrene by Psychrotrophic Antarctic Bacterium

**DOI:** 10.3390/polym15081841

**Published:** 2023-04-11

**Authors:** Pui Mun Tang, Syahir Habib, Mohd Yunus Abd Shukor, Siti Aisyah Alias, Jerzy Smykla, Nur Adeela Yasid

**Affiliations:** 1Department of Biochemistry, Faculty of Biotechnology and Biomolecular Sciences, Universiti Putra Malaysia, Serdang 43400, Malaysia; 2Institute of Ocean and Earth Sciences, C308 Institute of Postgraduate Studies, University of Malaya, Kuala Lumpur 50603, Malaysia; 3National Antarctic Research Centre, B303 Institute of Postgraduate Studies, University of Malaya, Kuala Lumpur 50603, Malaysia; 4Institute of Nature Conservation, Polish Academy of Sciences, Mickiewicza 33, 31-120 Kraków, Poland

**Keywords:** polystyrene microplastics utilisation, weight loss, additives, Antarctic soil, *Brevundimonas* sp.

## Abstract

Polystyrene (PS) and microplastic production pose persistent threats to the ecosystem. Even the pristine Antarctic, which is widely believed to be pollution-free, was also affected by the presence of microplastics. Therefore, it is important to comprehend the extent to which biological agents such as bacteria utilise PS microplastics as a carbon source. In this study, four soil bacteria from Greenwich Island, Antarctica, were isolated. A preliminary screening of the isolates for PS microplastics utilisation in the Bushnell Haas broth was conducted with the shake-flask method. The isolate AYDL1 identified as *Brevundimonas* sp. was found to be the most efficient in utilising PS microplastics. An assay on PS microplastics utilisation showed that the strain AYDL1 tolerated PS microplastics well under prolonged exposure with a weight loss percentage of 19.3% after the first interval (10 days of incubation). Infrared spectroscopy showed that the bacteria altered the chemical structure of PS while a deformation of the surface morphology of PS microplastics was observed via scanning electron microscopy after being incubated for 40 days. The obtained results may essentially indicate the utilisation of liable polymer additives or “leachates” and thus, validate the mechanistic approach for a typical initiation process of PS microplastics biodeterioration by the bacteria (AYDL1)—the biotic process.

## 1. Introduction

Various plastics have been manufactured, marketed, and used since the 1950s as their functional features facilitate many human activities. Plastic is commonly described as a lightweight, inexpensive, durable, and robust material that is practical and easy to mould into any shape [1]. According to Chen et al. [2], plastic production hit 359 million tonnes (Mt) in 2018 and is projected to reach beyond 800 Mt by 2040. The significant growth of plastic usage and low recycling rate increases the plastic waste accumulation in the environment, thus causing plastic waste to be a global issue [3].

Plastics can break down abiotically into microplastics when exposed to high temperatures, ultraviolet light, and natural mechanical forces [4]. These processes are usually the initiators in plastic degradation such as cracking, cleavage, or oxidation of the chemical bonds create smaller particles, known as microplastics with a larger surface area and higher hydrophilicity that enhance microbial activity [3]. On the other hand, the biotic process of plastic degradation involves microorganisms such as bacteria, fungi, and algae to interact and alter the properties of polymers [5].

Polystyrene (PS) is an aromatic hydrocarbon synthesised from styrene monomers through polymerisation, resulting in a molecule with a high molecular weight [6]. PS is one of the thermoplastic polymers widely used in many industries due to its light weight, thermal and moisture resistance, long-lasting quality, and adaptability [7]. Like any other plastics, PS can be destructed into microplastics (<5 mm) in the environment via photodegradation and weathering [3]. The increased surface area of the particles enhances microbial attachment and, subsequently, PS deterioration [4]. Generally, the mechanisms involve biofilm formation with bacteria colonising the polymer surface, followed by bio-fragmentation of the polymer by extracellular enzymes, and lastly, bio-assimilation of the fragments as a carbon source to support bacterial growth [8]. For example, *Bacillus cereus*, *Bacillus gottheilii* [9], *Bacillus paralicheniformis* [10], and *Rhodococcus ruber* [11] are some investigated bacteria that have been shown to deteriorate PS microplastics by utilising them as a carbon source. The utilisation will eventually result in the reduction in microplastics’ weight, and simultaneously alter the chemical structure as well as the surface morphology of PS [12].

The presence and pollution of PS have been reported in different soil environments such as farmlands [13], floodplains [14], industrial areas [15], and coastal areas [16]. Moreover, plastic particles were also found to pollute remote and pristine regions such as the Antarctic through natural and human activities [17]. For example, Isobe et al. [18] detected PS microplastics in the Southern Ocean, while Bergami et al. [19] discovered traces of PS in the collembolan *Cryptopygus antarcticus*. As there is no report of any actual anthropogenic system applied to remove microplastics in natural environments, it is urgent to know the consequences of their interactions with microorganisms, both on their degradation rate and the ecosystems. Moreover, little is known of the effects they have when interacting with psychrotrophic bacteria inhabiting the pristine regions, with only a single study having been reported to date [20]. However, the extent of the utilisation of the microplastic was only determined based on weight loss and infrared spectroscopy analysis. No comprehensive visible inspection, which can be provided by a scanning electron microscopy, was used to visualise any notion of the biodeterioration process. In this study, we attempt to examine the Antarctic bacteria isolated from the soil sample in the utilisation of PS microplastics as their sole carbon source. The utilisation of PS was determined via the weight loss and reduction rate after incubation with the bacteria. Furthermore, the structural changes of the PS before and after the infusion were assessed through infrared (FTIR) spectroscopy and scanning electron microscopy (SEM).

## 2. Materials and Methods

### 2.1. Preparation of Polystyrene (PS) Microplastics

PS microplastics were prepared by grating the commercial PS product (Greiner Bio-One GmbH, Frickenhausen, Germany; Lot No: E10090MX) obtained from the plastic-producing industry using a steel file. The grated plastics were then filtered through a sieve with a mesh size of 0.3 mm. The filtered particles (<0.3 mm) were collected and sterilised by washing with 70% ethanol, followed by oven-drying at 60 °C for three days.

### 2.2. Isolation of the Antarctic Bacteria

Soil samples were aseptically collected from Greenwich Island, Antarctica (62°26′51.0″ S, 59°44′12.4″ W) (Figure 1), and maintained at −20 °C until further use. Five grams of the soil sample was suspended in 50 mL of sterile saline water (0.9% *w*/*v* NaCl). The solution was shaken for 3 h (15 ± 2 °C, 150 rpm) using an incubator shaker (LM-510RD, Yihder, Taipei, Taiwan) and was subjected to serial dilution up to 10^−7^. A total of 0.1 mL of the aliquot (10^−3^–10^−6^) was spread aseptically on freshly prepared nutrient agar (NA) (Oxoid Ltd., Cheshire, UK). The agar plates were incubated at 15 ± 2 °C for 12–14 days. Distinct colonies that developed were isolated and further purified using the streak plate method. The colonial morphologies of the isolates were observed and recorded.

### 2.3. Preliminary Screening of Isolates for PS Microplastics Utilisation

Growth profiles of the isolates were generated before the screening process. For the assessment of bacterial growth and microplastics utilisation, Bushnell Haas (BH) broth (HiMedia, Mumbai, India) consisting of MgSO_4_ (0.2 g/L), CaCl_2_ (0.02 g/L), K_2_HPO_4_ (1.0 g/L), KH_2_PO_4_ (1.0 g/L), NH_4_NO_3_ (1.0 g/L), and FeCl_3_ (0.05 g/L) was used throughout the experiment as the medium contains no carbon sources for the bacterial growth. Microbial inoculums were prepared by growing the isolates in nutrient broth (NB) (Oxoid Ltd., Cheshire, UK) at 17 ± 2 °C until the log phase in an incubator shaker (150 rpm). Then, the screening was performed by inoculating 10% of the bacterial cultures (OD_600_ = 0.7) into 45 mL of BH media supplemented with 0.2 g of PS microplastics as the carbon source. The flasks were shaken for 10 days in an incubator shaker (17 ± 2 °C, 150 rpm). Control sets were maintained with (i) the uninoculated medium with microplastics, and (ii) the inoculated medium without microplastics. The weight loss of PS microplastics was measured and calculated at the end of the incubation. All experiments were carried out in replicates [20,21].

#### 2.3.1. Determination of Weight Loss of Residual PS Microplastics

PS microplastics that remained in the liquid medium were retrieved through filtration (filter paper with a pore size of 11 µm), washed with 70% ethanol, and oven-dried (60 °C) overnight. The initial and final weight of PS microplastics were measured using an analytical balance (Mettler Toledo AL204) with a readability of 0.001 g. The weight loss percentage of PS microplastics was calculated using Equation (1):Weight loss percentage, % = (*W*_0_ − *W*/*W*_0_) × 100,(1)
where *W*_0_ is the initial weight of PS microplastics (g) and *W* is the final weight of PS microplastics (g).

### 2.4. Strain Identification

The screened isolates were identified via morphological (Gram staining) and molecular methods (16S rRNA gene sequencing). For the latter, genomic DNA was extracted using the DNeasy Blood & Tissue Kit (Qiagen, Germany) based on the provided protocol. Amplification of the 16S rRNA gene was performed using the universal primers: 27 F (5′-AGA GTT TGA TCC TGG CTC AG-3′) and 1492R (5′-TAC GGT TAC CTT GTT ACG ACT T-3′) with a gradient thermocycler (Hercuvan, Cambridge, UK). A final volume of 25 µL PCR reagent consisting of 1 µL DNA template, 1 µL of forward and reverse primers each (0.4 mM), 12.5 µL REDiant 2× PCR Master Mix, and 9.5 µL sterile deionised water was used. The PCR was carried out with the following conditions: initial denaturation at 94 °C for 3 min; 29 cycles of denaturation at 94 °C for 1 min; annealing at 58 °C for 1 min; extension at 72 °C for 2 min, and a final extension at 72 °C for 10 min with incubation at 4 °C. Purification of the PCR products was then performed using the EZ-10 Spin Column Purification Kits (BioBasic, Markham, ON, Canada) to remove unwanted nucleotides, oligonucleotides (<40-mer), enzymes, mineral oil, and other impurities. Quantification and qualification of the purified PCR products were analysed on 1% (*w*/*v*) agarose gel electrophoresis. Lastly, the purified PCR products were sequenced using ABI PRISM 3730xl Genetic Analyzer (Thermo Fisher Scientific(Applied Biosystems), Waltham, MA, USA).

### 2.5. Sequence and Phylogenetic Analysis

The obtained partial 16S rRNA gene sequences were identified using BLASTn [22]. Multiple sequence alignment was then used to construct the phylogenetic tree via the neighbour-joining method [23] with the MEGA11.0 software. *Staphylococcus aureus* strain MVF-7 and *Escherichia coli* strain NBRC 102203 were used as the outgroups for AYDL1 and AYDL2, respectively. The evolutionary distances were computed using the Jukes–Cantor model [24], while the robustness of the inferred tree was evaluated by bootstrapping with 1000 replicates [25].

### 2.6. Assay for PS Microplastics Utilisation

The assay was set up according to the method mentioned in Section 2.3 by using AYDL1 (log CFU/mL = 4.0) as the experimental strain. In brief, 10% of the bacterial strain was inoculated into 45 mL of BH medium supplemented with 0.2 g of PS microplastics. The experiment was performed in replicates and the flasks were incubated up to 40 days, where the entire content of the flasks was sacrificed to obtain independent samples at 10-day intervals. The assessments such as microbial counts (CFU/mL) and weight loss percentage of residual PS microplastics were measured accordingly to steps mentioned in Section 2.3.1. Negative control was maintained by using an inoculated flask without microplastics.

### 2.7. Determination of Reduction Rate of PS Microplastics

The microplastics’ reduction rate was calculated by applying the first-order kinetic model shown under Equation (2):*K* = −1/*t* (ln(*W*/*W*_0_)),(2)
where *K* indicates the first-order rate constant for PS microplastics uptake per day, *t* indicates the time in days, *W* indicates the final weight of PS microplastics (g), and *W*_0_ indicates the initial weight of PS microplastics (g).

Then, the formula shown below was employed to calculate the half-life of PS microplastics by integrating the value generated from the removal rate constant (Equation (3)):*t*_1/2_ = ln (2)/*K*,(3)

### 2.8. Fourier Transform Infrared (FTIR) Analysis

The changes in chemical structure between the untreated and the treated PS microplastics were analysed by conducting the FTIR analysis via the attenuated total reflection (ATR) method. Treated PS microplastics that remained in the BH medium after 40 days of incubation were retrieved through filtration, washed with 70% ethanol, and oven-dried at 60 °C overnight. The ATR-FTIR analysis was carried out using the Spectrum 100 FTIR spectrometer (PerkinElmer, Shelton, CT, USA) with a frequency ranging between 4000 and 650 cm^−1^.

### 2.9. Scanning Electron Microscope (SEM) Analysis

The non-destructive microscope was used to examine changes in the surface morphology of the PS microplastics subjected to bacterial incubation. The treated PS microplastics were retrieved from the culture medium according to the steps in Section 2.8. The untreated and treated PS microplastics were dispersed evenly on the carbon-coated SEM stub. Then, all samples were sputter-coated with a gold layer at 20 mA under an argon (Ar) atmosphere (5 × 10^−2^ mbar) before visualising via SEM (JSM 6400 SEM, JEOL Ltd., Tokyo, Japan) with 1000× magnification.

### 2.10. Statistical Analysis

Data analysis was carried out using the non-parametric Mann–Whitney U test (SPSS software, v 26.0) to assess the weight loss of PS microplastics because a small sample size (*n* = 2) and two independent groups (AYDL1 and AYDL2) were used.

## 3. Results and Discussion

### 3.1. Isolation and Preliminary Screening of Antarctic Bacteria for PS Microplastics Utilisation

The Antarctic is a unique continent with limited flora and fauna but comprises abundant microbial diversity dominated by psychrophiles [26]. Cold-adapted bacteria have outstanding biotechnological benefits as living in an adverse environment prompts them to express remarkable features for survival. For example, they may have specific mechanisms in terms of metabolic regulation, cell membrane modification, or the synthesis of cold-active enzymes that help them to thrive in the cold [27]. Therefore, the Antarctic bacteria could potentially be agents that utilise PS microplastics for their growth while reducing the amount of microplastics.

This study successfully isolated four bacterial isolates exhibiting distinct colonial morphologies from the Antarctic soil samples. The bacteria displayed similar shapes and margins on the agar plates, yet the variations in pigmentation allowed for an easy characterisation between the isolates. Two out of four bacterial isolates labelled AYDL1 and AYDL2, which possessed a higher growth rate, were selected for the screening of PS microplastic utilisation in the BH medium. The reduction in terms of the PS microplastics’ weight in both bacterial cultures implies that both isolates could be potential agents that utilise PS microplastics (Table 1). According to Sanin et al. [28], carbon starvation tends to increase cell surface hydrophobicity due to the excess protein formation in the extracellular polymer matrix (EPS). This will in turn enhance the hydrophobic interaction and the affinity of bacterial cells for PS as the carbon source.

Isolate AYDL1 exhibited a better utilisation of PS microplastics as it recorded a higher weight loss percentage (*Mdn* = 18.5, *n* = 2) than AYDL2 (*Mdn* = 12.75, *n* = 2) (*U* = 0.000, *p* = 0.333, *r* = 0.7745), while the control exhibited no reduction in the PS microplastics’ weight). According to the statistical analysis, the difference in terms of weight loss percentage of PS microplastics between AYDL1 and AYDL2 is not as significant as *p* > 0.05. Gómez-de-Mariscal et al. [29] mentioned that the *p*-value in the Mann–Whitney *U*-test has an inverse correlation with the sample size where the *p*-value tends to decrease with the increase in sample size (*n*). Therefore, the small sample size (*n* = 2) used in this experiment could have weakened the statistical power of the analysis and subsequently given rise to a non-significance outcome. By this means, using a larger sample size would be necessary to verify the significance of the findings.

Furthermore, some of the PS microplastics were observed to settle down to the bottom of the BH medium for both isolates after 10 days of incubation. This phenomenon coincides with the finding of Auta et al. [9] where the increased density may contribute to the sinking of microplastics on the fifth day of incubation due to the adherence and colonisation of bacteria on the polystyrene particles. Moreover, both cultures were observed to be more turbid after the incubation, indicating that the bacteria could utilise PS as a carbon source to support their growth.

### 3.2. Identification of PS-Utilising Strain

Isolate AYDL1 is characterised as a Gram-negative rod bacterium while AYDL2 is a cocci-shaped cell that displayed a positive Gram’s reaction. Both isolates formed circular and smooth colonies but expressed different pigmentation on nutrient agar (NA). Moreover, 16S rRNA gene sequence analysis was used as a molecular approach to identify the bacteria. From the sequencing result, the partial 16S rRNA sequence lengths of AYDL1 and AYDL2 were 1348 bp and 1395 bp, respectively. The nucleotide blast analysis showed that strain AYDL1 is a member of *Brevundimonas* sp. (>96% identity). While AYDL2 is revealed to be closely related to several species belonging to the order of Micrococcales in the phylum Actinomycetota, the strain exhibited the highest similarity percentage (99.43%) with *Dermacoccus* sp. against the other species such as *Austwickia* sp., *Luteipulveratus* sp., *Calidifontibacter* sp., *Yimella* sp., *Orthinimicrobium* sp., *Kineosphaera* sp., *Rudaeicoccus* sp., *Janibacter* sp., *Mobilicoccus* sp., and *Intrasporangium* sp. Therefore, AYDL2 is assigned as *Dermacoccus* sp.

Based on the partial 16S rRNA sequences, a neighbour-joining phylogeny tree was constructed by using *Staphylococcus aureus* strain MVF-7 and *Escherichia coli* strain NBRC 102203 as the outgroups for AYDL1 and AYDL2, respectively. A close relationship was shown between AYDL1 and *Brevundimonas albigilva* strain NHI-13 as they are placed within the same clade (Figure 2). A bootstrap value of 74% indicated that the clade at a deeper node was well-supported. On the other hand, the clade with strain AYDL2 and *Dermacoccus nishinomyaensis* strain DSM 20448 was supported with significant bootstrap value inferring the close evolutionary relationship between these two strains (Figure 3).

### 3.3. Assay of Polystyrene (PS) Microplastics Utilisation by AYDL1

The genus *Brevundimonas* is commonly attributed as a hydrocarbon degrader that has been shown to utilise compounds such as *n*-alkanes, polycyclic aromatic hydrocarbons (PAH) [30], diesel oil [31], and PS [32]. Moreover, *Brevundimonas* sp. was also found in the hydrocarbon-contaminated soils from Ross Island, Antarctica [33]. These findings corroborate the ability of the psychotrophic *Brevundimonas* sp. to be a good candidate for PS microplastics utilisation.

To further understand the PS microplastics utilisation by the Antarctic bacteria, *Brevundimonas* sp. strain AYDL1 was selected for the assay of PS microplastics utilisation because it exhibited a better activity in terms of the PS microplastics weight loss percentage during the preliminary screening (Table 1). The assay on PS microplastics utilisation showed that the strain AYDL1 tolerated PS microplastics well under prolonged exposure with a weight loss percentage of 19.3% after 10 days of incubation (Figure 4). According to Friedrich et al. [34], *Brevundimonas* sp. is a *K*-strategist that reproduces slowly but maintains a stable existence by adapting to the nutrient-poor environment, which in this case, corresponds to adapting to and utilising the PS microplastics as the carbon source. Furthermore, Mor and Sivan [11] mentioned that carbon starvation may promote the attachment of bacterial cells to the PS surface and result in biofilm formation, a mechanism that initiates the utilisation. This is because the biofilm creates a microenvironment that favours the interaction between the cells and the PS microplastics, allowing the bacteria to utilise PS particles as a carbon source for their survival [35].

The data demonstrate that AYDL1 recorded the highest removal rate of 0.0211 day^−1^ and the shortest half-life of approximately 33 days on the tenth day with 19.3% of weight loss (Table 2). This may imply that the bacteria and microplastics had the most interaction around the tenth day of incubation. Furthermore, the higher removal rate and shorter half-life compared to previous studies [36,37] are believed to be influenced by the smaller weight (0.2 g) of PS microplastics used in the experiment. However, an unfavourable environment resulting from the build-up of toxic wastes or condition shifts in the medium can also affect bacterial growth and disrupt the overall performance of strain AYDL1 in utilising and removing PS microplastics after the tenth day [9]. No contamination was observed in the negative control flasks from day 0 (4.15 ± 0.21 log CFU/mL) to day 40 (0.76 ± 1.0 log CFU/mL). The complete exhaustion of carbon as a source of growth for bacteria at the end of the experiment may cause cell death, and thus lead to growth reduction.

Nonetheless, the substantially analogous weight loss percentage and microbial counts were observed in each of the intervals throughout the experiment. The results may imply the utilisation of the smaller polymer monomers (styrene) or the plastic additives—a series of small molecules that are added to the polymer to aid the manufacturing processes of the final plastic products [38,39,40,41]. The near-constant percentage of weight loss of ≤20%—one of the criteria often considered as a standard for the confirmation of plastic polymer degradation—may also corroborate the potential utilisation of the said additives [41].

Table 3 compares the weight loss of PS microplastics utilised by various types of bacteria under different parameters. Mor and Sivan [11] highlighted the importance of biofilm formation in initiating biodegradation by *Rhodococcus ruber* which had reduced 0.8% of pure PS flakes in 8 weeks. *Bacillus* spp. and *Pseudomonas* spp. are known to be versatile bacteria in the bioremediation field. With regard to PS, *Bacillus subtilis* was reported to degrade and reduce the weight of commercial PS by 58.82% [37]. Auta et al. [9], who similarly used commercial PS and treated the microplastics with ultraviolet (UV) light, showed that *Bacillus cereus* and *Bacillus gottheilli* recorded 7.4% and 5.8% of weight loss in PS, respectively. On the other hand, *Pseudomonas aeruginosa* was able to degrade PS with a weight loss percentage of 63.43% [42], while 1.45% of PS microplastics were reduced by *Pseudomonas lini* in 30 days of incubation [43]. Previous studies conducted on the degradation of high-impact PS (HIPS), a type of polybutadiene-contained thermoplastics, disclosed that *Brevundimonas* sp. from India degraded about 3% of the polymer’s weight [44]. In contrast to this study, *Brevundimonas* sp. AYDL1 utilised and reduced the PS microplastic’s weight by 18% in 40 days. Despite the higher temperature used in the previous study, AYDL1 demonstrated a higher percentage of weight loss, corresponding to a higher rate of PS utilisation. This infers that different types and compositions of polystyrene may affect the efficiency of a bacterium in utilising microplastics.

### 3.4. FTIR Analysis of PS Microplastics

The utilisation of PS microplastics by bacteria is likely to involve changes in the chemical bonds. The infrared spectroscopy with a frequency ranging between 4000 and 650 cm^−1^ was conducted to compare the variation in chemical structures between the untreated (control) and treated PS microplastics. Figure 5 showed that the obtained ATR-FTIR spectrum of the untreated PS microplastics (control) differs from the spectrum of pure PS used in the studies by Arsalan and Rafiuddin [47] and Chaudhary et al. [48]. The control had a similar C-H stretching of aliphatic groups to the pure PS at the bands of 3026 and 2925 cm^−1^. Moreover, the sharp peaks of alkane bend for methylene and methyl at 1493 and 1450 cm^−1^, respectively, were also observed. The presence of a benzene ring was shown with the peak appearing at 1600 cm^−1^. However, several characteristic peaks were observed only in the untreated PS microplastics. For example, the respective absorption peaks at 1740, 1368, and 1218 cm^−1^ responsible for the C=O stretching, C-F stretching, and C-O stretching, respectively, were rarely observed in the pure PS. This indicates that the commercial PS product used may be a pre-oxidated polymer and/or possess distinct functional groups due to the incorporation of polymer additives during the production [49]. For instance, a review by Yeung et al. [50] mentioned that PS can be subjected to fluorination to increase water repellence of the polymeric materials while the carbonyl group can be introduced to PS through acylation to improve thermal and corrosion resistance.

The chemical structures of PS microplastics were altered after 40 days of incubation with AYDL1. The alteration of the C=O stretch at 1740 cm^−1^ to an inverted band is the most evident change that occurred. Similarly, Ahmad Tarmizi and Kasmuri [51] observed that the C=O group in the PS microplastics vanished after 21 days of incubation with the bacteria. This may be due to the utilisation of the carbonyl groups in the metabolism pathways such as the tricarboxylic acid (TCA) cycle for energy production [52]. Moreover, the strong peak for C-O stretching in the untreated PS microplastics disappeared but an inverted peak was observed. According to Brandt et al. [53], poor background correction could be a factor leading to the appearance of an inverted band. Furthermore, a new band corresponding to the phenolic O-H bond was observed at 1322 cm^−1^. All characteristic peaks in the FTIR spectrum were observed to have increased the transmittance percentage after the incubation with AYDL1. Elevated transmittance values observed at 3028 and 2922 cm^−1^ could indicate that the C-H alkane chain in PS microplastics was cleaved and utilised by the bacteria as a carbon source.

### 3.5. SEM Analysis of PS Microplastics

SEM was used to characterise the changes in the surface morphology of PS microplastics after incubation with strain AYDL1. As shown in Figure 6a, the untreated PS microplastics had smooth and intact surfaces without any noticeable deformed structures. In contrast, the PS microplastics treated for 40 days were damaged as a more porous and rougher surface was observed on the particles (Figure 6b). This demonstrates that strain AYDL1 utilised PS microplastics and at the same time destructed the surface morphology of the polymer. However, in light of the findings in both the assay and the infrared spectroscopic analysis, the rough surfaces on the treated sample may also signify the initial phase of cracking, which encourages microbial attachment and the subsequent prompting of biofilm synthesis.

According to Hou and Majumber [54], the release of extracellular enzymes by bacteria could be one of the factors contributing to the alteration in the surface morphology of the PS microplastics [48]. Moreover, according to Hidayat et al. [55], who noticed damages on PS microplastics after 3 weeks of incubation, the deformation will be more visible and evident with a longer incubation period. Other studies conducted on the PS microplastics utilisation by bacteria also reported similar results on the structural deformation after 30 days of incubation [42,43,56].

## 4. Conclusions

This study discloses the extent to which psychotrophic bacteria utilise PS microplastics. Among the investigated isolates, *Brevundimonas* sp. AYDL1 exhibited the most efficient utilisation of PS microplastics. This study increases the apprehension of bacterial ability to utilise PS microplastics as a carbon source. This was established by monitoring the growth of the bacterial isolate in the medium supplemented with PS microplastics as well as characterising the PS particles in terms of weight loss; reduction rate; half-life; formation and disappearance of functional groups; and surface modification. Even though both the utilisation assay of PS microplastics and the structural analysis of PS microplastics imply a positive utilisation of the polymers, the monotonous pattern of the PS utilisation assay may suggest that the bacterial strain (AYDL1) utilised the polymer additives as the carbon source, instead of the polymer itself. The presence of several uncommon characteristic peaks in the untreated (control) sample at 1740, 1368, and 1218 cm^−1^, which often denotes the C=O stretching, C-F stretching, and C-O stretching, respectively, strongly implies that the commercial PS product may be a pre-oxidated polymer that has been incorporated with plastic additives during the plastic manufacturing processes. The intention to utilise the commercial PS material in this project is to mirror the actual wastes heavily incorporated with various additives that are often found in most plastic-polluted areas. By incorporating the commercial PS materials in this study, new insights into the utilisation of “tailored” PS by the bacterial strain could be attained.

In order to develop a more comprehensive practice and workflow for the microplastic biodeterioration and biodegradation studies, future studies should commence with the characterisation of a plastic material used in the proposed study. This is to substantiate an absolute understanding for the differentiation of the degradation of either the polymer and/or the corresponding plastic additives. The validity of microplastic degradation or utilisation should also depend profoundly on the changes of the polymer structure, the physical loss of the plastic mass, as well as the generation of the plastic metabolites. Credible evidence of plastic biodegradation may likely be achieved using a combination of practices from all three groups. Future studies can also focus on the elucidation of possible enzymes, such as hydroxylase or monooxygenase, to improve the comprehension of the bacterial activity as well as discern the anaerobic or aerobic pathways of PS microplastics utilisation. This could be achieved with the application of bioinformatics, transcriptome, and metabolomics—advanced analytical tools used comprehensively in different studies. Moreover, for smaller subsets of studies, optimisation testing on the parameters such as incubation period, temperature, and addition of enzymes or substrates could also help to create an optimised system that can be applied in the environment to reduce the number of PS microplastics in the future.

## Figures and Tables

**Figure 1 polymers-15-01841-f001:**
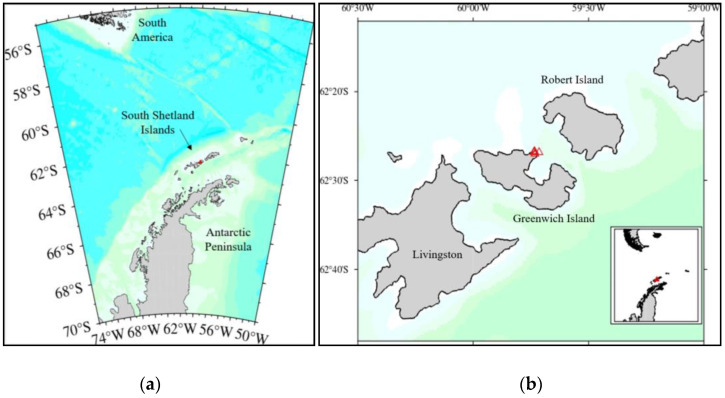
Map of the study area. (**a**) Location of the South Shetland Islands in relation to South America and the Antarctic Peninsula, (**b**) Greenwich Island within the South Shetland Islands which includes the sampling location (**∆**).

**Figure 2 polymers-15-01841-f002:**
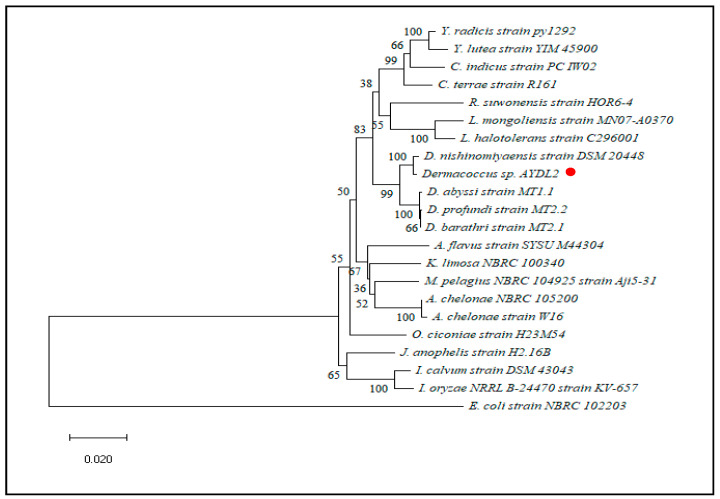
Phylogenetic tree of partial 16S rRNA gene sequences using the neighbour-joining method for AYDL1 (●). (GenBank: OQ135211).

**Figure 3 polymers-15-01841-f003:**
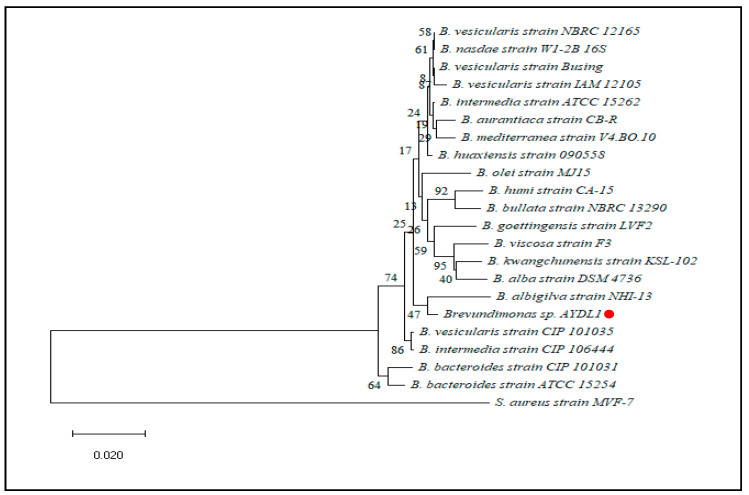
Phylogenetic tree of partial 16S rRNA gene sequences using the neighbour-joining method for AYDL2 (●). (GenBank: OQ135212).

**Figure 4 polymers-15-01841-f004:**
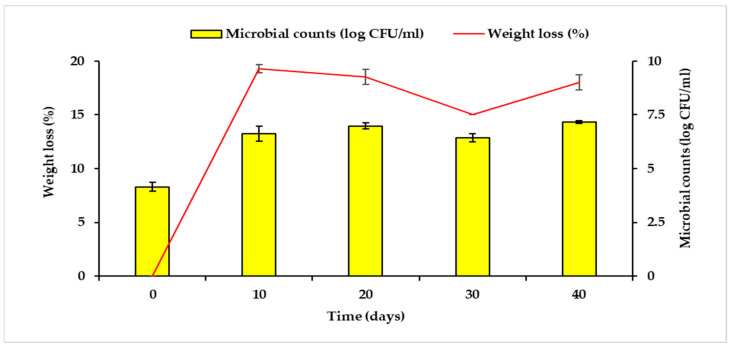
Weight loss percentage of the PS microplastics and the microbial counts in Bushnell Haas (BH) medium inoculated with strain AYDL1. The experimental flasks were shaken for 40 days at 150 rpm and 17 ± 2 °C. Data represent mean ± standard deviation (SD), *n* = 2.

**Figure 5 polymers-15-01841-f005:**
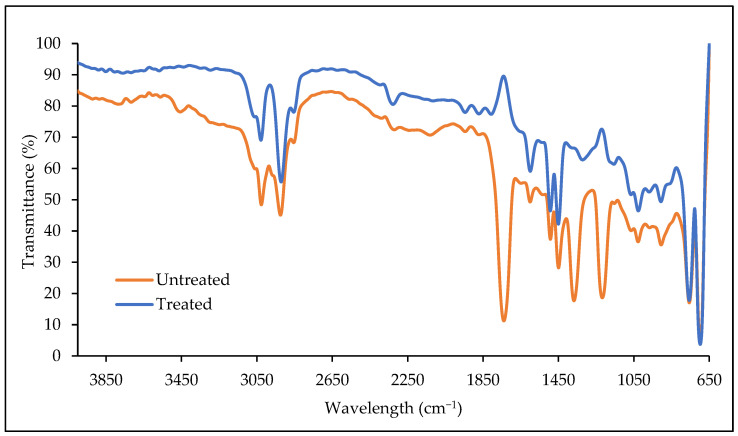
Fourier transform infrared (FTIR) spectra of untreated PS microplastics and treated PS microplastics after incubation with AYDL1 for 40 days in BH medium.

**Figure 6 polymers-15-01841-f006:**
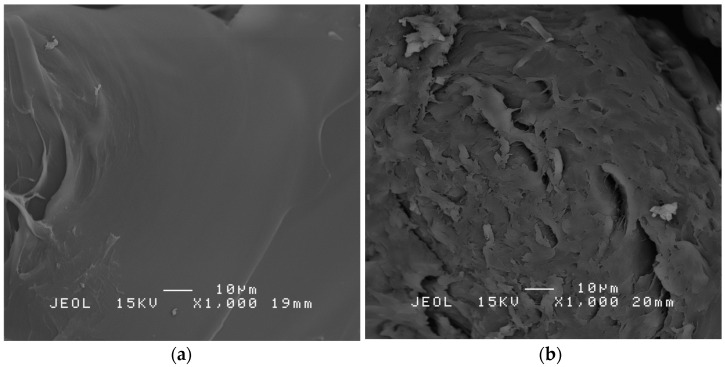
Scanning electron microscope (SEM) images of (**a**) untreated and (**b**) treated PS microplastics. The samples were viewed under SEM (JSM 6400 SEM, JEOL Ltd., Tokyo, Japan) with 1000× magnification.

**Table 1 polymers-15-01841-t001:** Weight loss percentage of polystyrene (PS) microplastics in uninoculated negative control and the cultures of isolate AYDL1 and AYDL2.

Isolate	Initial Weight (g)	Final Weight (g)	Weight Loss (%)
Negative control	0.200	0.199 ± 0.001	0.750 ± 0.354
AYDL1	0.200	0.163 ± 0.003	18.50 ± 1.414
AYDL2	0.200	0.175 ± 0.002	12.75 ± 1.061

**Table 2 polymers-15-01841-t002:** Removal rate constant and half-life of PS microplastics in the culture of strain AYDL1.

Day	Initial Weight (g)	Final Weight (g)	Weight Loss (%)	Removal Rate Constant, K (Day^−1^)	Half-Life (Days)
0	0.2	0.2	0	0	∞
10	0.2	0.162 ± 0.001	19.3 ± 0.4	0.0211	33
20	0.2	0.163 ± 0.001	18.5 ± 0.7	0.0102	68
30	0.2	0.170 ± 0.000	15.0 ± 0.0	0.0054	128
40	0.2	0.164 ± 0.001	18.0 ± 0.7	0.0050	137

**Table 3 polymers-15-01841-t003:** Previous and current studies of PS microplastic utilisation by bacteria.

Bacterial Strain(s)	Origin	Type of PS	Incubation Period	Temperature (°C)	Weight Loss (%)	Reference(s)
*Rhodococcus ruber*	Agricultural soil, Israel	Pure PS flakes	8 weeks	35	0.8	[11]
*Bacillus subtilis*	Various soils, India	Commercial PS in strips (0.5 × 1 cm)	1 month	RT ^1^	58.82	[37]
*Staphylococcus aureus*					37.50	
*Streptococcus pyogenes*					11.11	
*Pseudomonas aeruginosa*	n.d ^2^	Styrofoam solution	30 days	25	63.43	[42]
*Bacillus* spp.	Soil from plastic dumping yard, India	High-impact PS (HIPS) film (1 × 1 cm)		30	23.70	[45]
*Pseudomonas* spp.					<10	
*Enterobacter* sp.					<4.0	[44]
*Brevundimonas* sp.					<3.0	
*Acinetobacter johnsonii*	Soil from a trash-containing dump, Republic of Korea	PS powder		28	1.52	[43]
*Pseudomonas lini*					1.45	
*Bacillus cereus*	Mangrove soil, Malaysia	Commercial PS Treated with UV light for 25 days	40 days	29	7.4	[9]
*Bacillus gottheilii*					5.8	
*Exiguobacterium* sp. strain YT2	Gut of mealworms (*Tenebrio molitor*)	Styrofoam film (50 × 50 mm)	60 days	22–24	7.4 ± 0.4	[46]
*Brevundimonas* sp. AYDL1	Soil, Antarctic	Commercial PS (<0.3 mm)	40 days	18	19.3	This study

^1.^ room temperature; ^2^ not determined.

## Data Availability

All data are provided within the manuscript.

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
