# Peer review of "Evaluation of the Deterioration of Untreated Commercial Polystyrene by Psychrotrophic Antarctic Bacterium"

_polymers, 2023, doi:10.3390/polym15081841_

Round 1

Reviewer 1 Report

The use of psychotropic bacteria in plastic degradation seems like a promising avenue for future research. Both the methodology and the results are strongly backed up by the data. The discussions are adequately supported by the findings, and I believe that this manuscript can be published with only a few modifications required.

1. Did the authors perform the experiment with the bacteria on their own as a form of negative control? If so, I would like to see the growth of the bacteria in the same BH medium but without the presence of any plastics.  If this is the case, the authors need to input their findings and then discuss them.

2. The manuscript requires minimal elaboration and language edit, particularly in the introductory and concluding sections.

3. There are a few typos in the manuscript; please correct them.

If the authors are interested in using this information in future research, I have some suggestions for them.

1. It is recommended to have the positive control of the organism when comparing the use of plastics as carbon source. For instance, to make a fair comparison, I would have used the same Brevundimonas bacteria (either a type strain or other isolates from the culture collection center).

2. The experiment was carried out under a single set of temperature conditions, despite the fact that the strains were obtained from the antarctic. It should be noted that the utilization performances of the strains were not tested under different temperature regimes. Optimal conditions for improved use could have been determined with further testing at varying temperatures.

3. As a final step, the authors should consider conducting molecular expression studies in order to identify the principal genes contributing to the degradation of the plastic.

Author Response

Thank you for the constructive review. We have made all the adjustments and provided sufficient proof as requested. We hope this paper can be published in Polymers. Thank you. 

Reviewer 2 Report

The manuscript entitles “Evaluation of the Deterioration of Untreated Commercial Poly- 2 styrene by Psychrotrophic Antarctic Bacterium" by Pui Mun Tang, et al., explained about the biodegradation of polystyrene by soil bacteria from Greenwich Island, Antarctica, were isolated. There are some revisions for better understanding as below:

1.    In the introduction section add 3-4 previous literature indicating the limitation that you want to address in current studies.

2.    Clearly, indicate the novelty of the work in the manuscript.

3.    Add references in section 2.3.

4.    Perform AFM before and after with surface roughness and calculate it.

5.    Authors you perform experiments in replicates, yes/ no? provide images of the experimental setup.

6.    In Figure 4 weight loss and microbial count is confusing to change the style of the graph.

7.    Why did you observe weight loss in only 40 days?

8.    Improve the conclusion and give statistical values related to your results. Revise this part and write your outcome and future recommendations in one paragraph.

Author Response

(The authors gave the same response as above.)

Reviewer 3 Report

The manuscript “Evaluation of the Deterioration of Untreated Commercial Polystyrene by Psychrotrophic Antarctic Bacterium” from Tang et al. deals with the identification of a couple of new Antarctic bacterial strains that can growth on PS microplastics from a commercial product and determine a partial weight loss of 18%, even at relatively low temperatures.

Although the data presented are in accordance with the general claim, substantial revision to the text is needed to change the overall conclusion of the work, including a clarification on the experiments. Indeed, the presented evidence goes in the direction of the degradation of the pre-oxidized polymer components or additives contained in the PS microplastics and the degradation stops after just 10 days (there is not an increasing amount of weight loss from 10 to 40 days), while the authors did not check what happened  between day 0 and 10. These changes should be also reflected in the abstract section.

In the following the list of points to address throughout the manuscript:

- L. 42 Fix the text: “create microplastics which larger surface area and…“ which -> with

- L. 52-53: it is suggested to explicit the extent of PS waste that is actually not recycled, to improve in understanding the impact of finding PS degrading systems for bioremediation, as too often this is not clear from the introduction section of recalcitrant petroleum-based plastic biodegradation manuscripts, making difficult for readers (both academic, private or public administration) to evaluate the extent of the problem, which is crucial to be understood for a matter as plastic accumulation in the environment. In this regard, please refer to https://doi.org/10.3390/ijms24043877 (Fig. 6)  in which it is reported that unrecycled PS waste forms only 2% of the total unrecycled plastic waste (macro + microplastics); moreover, the actual systems of biodegradation are poorly effective on PS and mainly observed to be active on pretreated PS materials or co-polymerized additives. Therefore, in the introduction section the authors should put less attention on the problem of the PS microplastics in the natural surroundings (i. Determine the effective amount of microplastics in natural environments is challenging and can be controlled only in composting sites and artificially altered environments ii. Biodegradation of PS microplastics was always shown to be incomplete, even in presence of pre-treatments, including the examples reported by authors in Table 3), and instead focus the rationale of the work on understanding the bacterial strains are more prone to cause some degradation of such recalcitrant plastics and, eventually, the mechanisms by which this happens.

- To prove any biodegradation of the PS polymer inside the microplastics, the authors should measure the degradation products (soluble or in suspension) in the culture medium. It could be that the microorganisms are mainly causing a mechanical disruption during growth, by degrading the additives that are the assimilated as carbon source

 - Line 78: As the polymer details were not given, it is fundamental to add a characterization of the material, or ask to the producer what the characteristics of the source PS material are (purity, presence of industrial treatments such as bromination, other additives, etc.): organisms were previously shown to prefer to biodegrade additives rather than the C-C backbone of PS polymers (10.1016/j.jhazmat.2020.123534)

 - Line 81: Why did the authors use this thermal pre-treatment for PS? Could this facilitate the biodegradation action? It is unlikely that in natural ecosystems microplastics go through this temperature, especially in Antarctic environments.

 - Line 111: Which is the filter size? It could be that the action of microorganisms to be mainly mechanical and generates particle with a size that is lower than the original one, and capable to pass through the filter under use. Therefore, it is critical to know this methodological detail.

 - Paragraph 2.7: The calculation of a reduction rate assumes that the process of PS microplastics degradation comes up with a constant rate, and up to complete or near complete biodegradation. Anyway, authors do not test such assumption, while previous studies have shown that untreated PS microplastic biodegradation stops at some point often after < 30% biodegradation, and the rate is varying during the microbial incubation. Therefore, the calculation of a reduction rate with the provided formula is misleading and should be removed from the manuscript.

 - Table 1: Report the error also for the weight loss.

 - Paragraph 3.1: Please provide explicit quantitative data on growth curves (OD600) during the incubation, as the extent of OD growth is critical for supporting if microbes are using PS microplastics as carbon source.

 - Paragraph 3.3: According to Fig. 5, the growth rate is zero and not “steady”, as declared in the text. This, in turn, is not in accordance with previously reported data on a hypothetic increase of the turbidity due to organism growth during the incubation with PS microplastics.

 - Line 252: this indicates that the organism can tolerate prolonged exposure to the absence of nutrients but does not support the use of PS as the carbon source for growth.

 - Line 258: Yes, but organisms can also use additives and their degradation may result in mainly mechanical damages to PS microplastics: this means that the filter size used to recapture the microplastics must be declared in the methods section, as it may be that the particle size is lowered and microplastics pass through, while they are not really "biodegraded", but mechanically fragmented by the microbial biofilm.

 - Figure 4: The bacterial growth does not seem to change significantly. Is Figure 4 the result of a single incubation over time (the same flask measured at 10, 20, 30, 40 days) or of independent samples taken at different time points (10,20,30,40)? Moreover, why no data on the microbial counts at the beginning of the incubation (time 0) is provided? If the number of cells at the beginning is superior, this means that the cells are not effectively growing or maintaining themself on PS microplastics. This experiment is not conclusive without adding the information at time 0, while increasing the inoculation time up to 40 days seem not necessary, as any degradation has stopped after 10 days.

 - Table 2 confirm the suspect from Figure 4. If new 0.2 g of PS microplastics were not added at each time point (this is not clear from the methods section), any biodegradation is finished after 10 days of incubation and the variation in the weight loss is purely an experimental measurement error. If this is the case, this experiment must be repeated with smaller incubation times (1, 2, 5, 10 days). This reviewer suspects that the PS microplastics were not added each time, because otherwise in table 3 it would be logic to indicate the weight loss as to be 18% in 10 days and not 40 days, or 72% in 40 days.

 - Line 305: This indicated that the polymer was pre-modified, and motivates a more detailed analysis of the source PS material  (to be presented as new data in this manuscript or as a reference to a product scheme from the producer).

 - Line 312: This is a strong indication that the isolated microorganisms are using additives or pre-oxidized components of the treated microplastics, and not the PS polymers.

 - Figure 5: the figure must be labelled to understand the connection between the different peaks and the observations made in the text. The results reported in this figure indicate untreated PS microplastics seem to be more oxidated than the microbially treated ones. This is a strong indication that microorganisms are using the pre-oxidated polymers or additives present in the PS polymer, and not the PS polymer.

 - Line 330: This may be due to release of C-H present in the oxidated additives that were used by the microorganisms.

 - Conclusions must be changed indicating that the obtained result is not sufficient to demonstrate the cleavage and utilization of PS polymers contained in the microplastics, but that probably the pre-oxidized polymers or additives are the carbon source of the isolated strains.

 - Line 358: A macroscopic solid commercial PS material does not resemble the quality of plastic waste, which is usually characterized but a superior mixing of polymers, especially at microscopic scale.

 - Lines 360-361: According to the data presented in this paper (Figure 5), these pre-oxidized additives are the most probably biodegraded components of PS microplastics, which cannot be further degraded after 10 days.

 - Lines 363-364: It is not advisable to add this as a possible pre-treatment to study, considering that microplastic bioremediation is planned to be performed in the environment (the very last sentence of this work) and these are cold-adapted microorganisms.

 - In Table 3 the degradation rate of the present work should be reported after 10 days, as from 10 to 40 days seems not to change (Figure 4).

Author Response

Thank you for the constructive review. We have made the adjustment and provided sufficient proof as requested. Thank you

Round 2

Reviewer 2 Report

The manuscript entitles “Evaluation of the Deterioration of Untreated Commercial Polystyrene by Psychrotrophic Antarctic Bacterium" by Pui Mun Tang, et al., explained the biodegradation of polystyrene by soil bacteria from Greenwich Island, Antarctica, isolated.

Some minor revisions are required for better understanding as below:

1.     Precise the abstract just gives, exclude extra explanation and add it in the result and discussion part.

2.     As written in the introduction mentioned about the chemical structure is changed, explain it in the experimental section.

3.     Revise the conclusion part and write it to the point.

Author Response

Dear reviewer, 

Thank you for the constructive review. Based on the comments, we have made the changes to the manuscript.

Reviewer 3 Report

The interpretation given by the authors to the presented results was updated as suggested.

There are some things remained to address:

- Lines 88-90. "The concerns about microplastic pollution in the region urge investigation of the potential of psychrotrophic bacteria for PS microplastic utilisation" I suppose the authors meant the "pristine region". Moreover, it is suggested to be more clear on the real world problem:  we cannot easly collect and remove microplastics already in the enviornment, so we need to know more what happens to them (and the timings) in these enviornments, while interacting both with abiotic and biotic factors. So I suggest the authors to rely on https://www.sciencedirect.com/science/article/pii/S2772577422000301 and https://doi.org/10.3390/ijms24043877 and change that sentence in "As there is not any anthropogenic system actually applied to remove microplastics in natural environments, it is urgent to know the consequences of their interactions with microorganisms, both on their degradation rate and the ecosystems; moreover, little is known on the effects they have when interacting with psychrotrophic bacteria inhabiting pristine regions, with only a single data ...". You should also provide a ref. for the "single data" that is cited.

- Lines 92 - 94: "However, the extent of utilisation of the microplastic was only determined based on weight loss and infrared spectroscopy analysis. No comprehensive visible inspection, that can be provided by a scanning electron microscopy, was used to visualize the biodeterioration." apply the fixes in italic and simplify the sentence

- Line 174: "...was performed in replicates and the flasks were incubated for 40 days..." Repalce per with up to

- Line 175: "..the entire content of flasks were sacrificed (independent samples) per 10 days.." Fix "were" with "was" and replace "per" with "to obtain independent samples at 10-day intervals"

- Lines 334-335: "Nonetheless, the significantly consistent weight loss percentage and microbial counts was observed in each of the interval throughout the experiment". As the authors collected independent samples after different time intervals, essentially this means that the weight loss is stopped after 10 days (Figure 4). It could be consistent between day 0 and 10, but no information is provided for shorter incubation intervals. Awaiting other studies, the comment on the significantly consistent weight loss has to be removed.

- Lines 428 - 430: The problem is not about the formula used to provide a kinetics of plastic degradation, which is right, but on the assumption that is required to provide a half-life estimate for this and any other process: the kinetics (parameter K calculated form eq. 1) has to be the same or similar from the beginning to the end of the considered process (I suppose complete biodegradation); from the data presented in this work, it is clear that the biodegradation process with the rate described by the authors comes to an end after 10 days and it is not known how it varies form day 0 to 10. So, it could be left a kinetic parameter (even if it may be not adequate because the rate may vary a lot in the first 10 days), but this must be related to the process studied in this manuscript (40 days) and not used to define a half-life estimate. Therefore, you have to change accordingly the text to adapt the degradation rate over the time of your experiments and not use a half-life of PS microplastics accoring to the defined rate, as the provided data conclude that the process is not proceeding in a constant rate, which is a necessary condition to define a half-life accoring to equation (2).

Author Response

Dear reviewer,

Thank you for the constructive review. We have made the changes based on the comments.
